# Exploring Gastrointestinal Health in Diabetic Cats: Insights from Owner Surveys, Ultrasound, and Histopathological Analysis

**DOI:** 10.3390/vetsci12060529

**Published:** 2025-05-29

**Authors:** Marisa Esteves-Monteiro, Cláudia S. Baptista, Diogo Cardoso-Coutinho, Clara Landolt, Patrícia Dias-Pereira, Margarida Duarte-Araújo

**Affiliations:** 1Associated Laboratory for Green Chemistry of the Network of Chemistry and Technology (LAQV@REQUIMTE), University of Porto (UP), 4050-313 Porto, Portugal; mmem89@gmail.com (M.E.-M.); pdiaspereira@yahoo.com.br (P.D.-P.); 2School of Medicine and Biomedical Sciences, University of Porto (ICBAS-UP), 4050-313 Porto, Portugal; up202109633@edu.icbas.up.pt; 3Laboratory of Pharmacology, Department of Drug Sciences, Faculty of Pharmacy, University of Porto (FFUP), 4050-313 Porto, Portugal; 4Department of Veterinary Clinics, ICBAS-UP, University of Porto, 4050-313 Porto, Portugal; 5Veterinary Hospital of the University of Porto (UPVet), 4050-313 Porto, Portugal; claralandolt@hotmail.com; 6AL4AnimalS, Center for the Study of Animal Science (CECA), 4099-002 Porto, Portugal; 7Department of Pathology and Molecular Immunology, ICBAS-UP, 4050-313 Porto, Portugal; 8Department of Immuno-Physiology and Pharmacology, ICBAS-UP, 4050-313 Porto, Portugal

**Keywords:** cats, diabetes, gastrointestinal tract, ultrasound, histopathology

## Abstract

Diabetes is a common disease in cats that can affect many parts of the body, but little is known about how it impacts the digestive system. In humans, diabetes often causes stomach and intestinal problems, yet similar effects in diabetic cats have not been well studied. This study looked at whether diabetic cats also experience digestive changes. The owners of diabetic cats answered a short questionnaire about any stomach or bowel issues they noticed. Then, each cat had an ultrasound scan of different parts of the digestive system. In some cats that had passed away, tissue samples were also studied under a microscope. The study included thirteen pet cats with diabetes but no previous digestive diseases. Most owners (83%) reported some digestive symptoms. All cats had thicker walls in parts of the stomach and small intestine. The tissue analysis confirmed changes throughout the digestive tract, including thicker muscle layers, inflammation, and more fibrous tissue. These results suggest that diabetic cats may have hidden digestive changes similar to those found in humans with diabetes. This knowledge can help veterinarians better understand and manage the full impact of diabetes in cats.

## 1. Introduction

Diabetes is a serious, chronic disorder [1] that is a major contributor to mortality and morbidity worldwide [2,3]. This disorder is one of the most common metabolic diseases in domestic pets, occurring in 0.21% to 1.24% of cats [4], with a higher prevalence described in Burmese cats [5]. Most cases of diabetes occur in middle-aged to older cats [6], with obese, neutered or not, males being more commonly affected than females [7]. Similar to human diabetes, this pathology in cats is associated with high levels of mortality and morbidity [8].

While diabetes includes various forms, the main are type 1 diabetes (T1D) and type 2 diabetes (T2D) [9]. T1D, which only accounts for 5–10% of all diabetes cases [10,11], is a chronic disease characterized by a complete absence of insulin secretion [12]. On the other hand, T2D is characterized by a combination of insulin and a relative deficiency of insulin production, making up 90 to 95% of all diabetes cases [10,13,14]. Although diabetic cats can present both forms of the disease, approximately 80% of them exhibit insulin-independent DM similar to T2D in humans [7]. Feline diabetes is a heterogeneous condition resulting from a combination of impaired insulin action in the liver, muscle, and adipose tissue (insulin resistance) and β-cell failure [7]. Among the contributing factors, obesity and physical inactivity coupled with the consumption of high-energy diets stands out as a significant risk factor, exerting a profound impact on T2D prevalence [15].

Diabetic gastrointestinal (GI) complications are highly prevalent in the human population and constitute a significant cause of morbidity, which influence the patients’ health status and quality of life [16,17,18]. However, awareness of these complications among physicians is often limited, with scant knowledge and treatment options available [19,20]. The entire length of the GI tract appears to be affected by diabetes, with alterations observed from the esophagus to the rectum [20,21]. The classic GI symptoms of diabetes include post-prandial fullness with nausea, vomiting, bloating, abdominal pain, diarrhea, and/or constipation [22], and the GI tract of diabetic laboratory animals also exhibits extensive remodeling [23,24]. These structural changes are indicative of underlying pathological processes that may affect the functionality and motility of the GI tract [24,25,26].

Considering the significance of GI complications of diabetes in humans, it should be expected to find similar reports of these complications in our diabetic pets. However, there are almost no reports of GI changes in diabetic dogs and cats [27]. Diabetes has been suggested as one of the possible causes of GI dysmotility in critically ill dogs and cats [28], and in a review of cases of diabetes seen at the Colorado State University Veterinary Teaching Hospital, about 38% of those dogs and 31% of cats also had GI disease [27].

Bearing in mind the similarities between feline diabetes and human T2D [29], as well as the lack of knowledge about GI complications in diabetic cats [27] and their potential impact on wellbeing, the aim of this study was to investigate whether diabetic cats exhibit GI alterations. To achieve this, we asked owners to respond to an anamnesis directed at the GI tract in order to find possible GI changes in their cats following diabetes diagnosis. Additionally, ultrasound examinations of the GI tract of diabetic cats were performed and histopathological evaluations were conducted on diabetic cats donated for post-mortem examination. The findings from this study are expected to shed light on the prevalence and nature of GI complications in diabetic cats, ultimately guiding better management and treatment strategies for these animals.

## 2. Materials and Methods

### 2.1. Study Population

All protocols were previously approved by a local animal welfare body (ORBEA ICBAS-UP Nº381/2020). Cats diagnosed with diabetes and receiving treatment at the Veterinary Hospital of the University of Porto (UPVet) from 2022 to 2025 were initially considered for the study (owners feedback and ultrasound evaluation). Subsequently, a rigorous selection process was implemented to ensure that the study’s results would accurately reflect only the impact of diabetes on the GI tract without confounding factors. Accordingly, all animals were subject to physical examination, and medical records were thoroughly reviewed. The exclusion criteria were the following: (i) pre-existing GI diseases, such as inflammatory bowel disease or GI neoplasia; (ii) signs of GI changes prior to the diagnosis of DM—such as vomiting, diarrhea, anorexia, or weight loss; (iii) previous treatments with corticosteroids, non-steroidal anti-inflammatory drugs, or antimicrobials within 30 days before undergoing the abdominal ultrasonographic examination [30]. After careful consideration, thirteen cats were selected for participation in the study. The sample size was calculated using the free software Sample Size Calculator (©2022—ClinCalc LLC, https://clincalc.com/stats/samplesize.aspx) to achieve 80% power and a significance level (*α*) of 5%, based on an expected 30% increase in diabetic cats compared to the general population.

For the histopathological analysis, six diabetic cats that died or were euthanized from causes unrelated to the GI tract and were donated to ICBAS-UP were included. Of these cats, only one had previously undergone an abdominal ultrasound. The clinicians that followed these animals were contacted to ensure that these cats had no history of GI disease and to evaluate their complete medical records. During the same period, nine non-diabetic cats that died from diseases not affecting the GI tract and free of GI lesions were randomly selected as controls. Animals showing GI alterations, such as the presence of parasites, were excluded during necropsy. These control cats were selected to closely match the diabetic cats in age and body weight, although it has already been shown that in adult cats, age, weight, or size does not affect the thickness of the GI tract [31]. The absence of GI disease was confirmed through histopathologic evaluation.

### 2.2. Owners’ Perception of Digestive Changes

The owners of the thirteen diabetic cats included in this study gave their informed consent and were asked to complete a simple yet comprehensive anamnesis directed to the GI tract. This survey, consisting of 27 questions with an estimated time of completion of 10 min, was meticulously designed to collect detailed information about any observed digestive changes following the diagnosis of diabetes. It covered the typical signs of diabetes (polydipsia, polyuria, polyphagia, and weight loss) and included specific questions about well-documented clinical signs of GI distress in cats, such as vomiting, diarrhea, and changes in appetite and bowel movements, which are widely cited in the literature as reliable indicators of gastrointestinal dysfunction in feline patients. A section was included at the end of the survey for owners to mention any additional GI-related observations they felt were relevant but were not specifically addressed in the previous questions.

### 2.3. Ultrasound Evaluation of the GI Tract

The selected diabetic cats underwent a comprehensive ultrasound evaluation of the entire GI tract performed by an experienced veterinary radiologist. The General Electric Logiq S8 R3 XDclear ultrasound machine by General Electric Healthcare, Carnaxide, Portugal was used to perform the examinations in the longitudinal and transverse planes using a 9–11 MHz linear probe. The gastric wall (including the rugal and inter-rugal folds), duodenum, middle jejunum, distal ileum, and distal colon walls were observed, with three separate ultrasound evaluations performed for each portion [31] for each cat. The results were compared to standard normal reference values for GI wall thickness as documented in the literature [32,33,34].

### 2.4. Necropsy and Histopathology

Both diabetic and non-diabetic cats selected as controls, which were donated for post-mortem necropsy, underwent a thorough examination, and samples of all portions of the GI tract were collected for histopathological evaluation. An experienced veterinarian pathologist performed the necropsies, with a special attention to the pancreas and GI tract. Photos of relevant lesions were taken, and a detailed necropsy report was prepared.

Samples (2 cm) of the stomach, proximal duodenum, middle jejunum, distal ileum, and distal colon were collected for histopathological analysis. These samples were routinely processed and paraffin-embedded, cut into 3 µm thick sections, and stained with hematoxylin-eosin (HE) for histological evaluation. Each section was examined under an optical microscope (Nikon, model Eclipse E600, Nikon Instruments, Miami, FL, USA) and photographed in four different representative regions using objective lenses of 2× and 4× (magnifications of 20× and 40×). The images were used to measure the thickness of the mucosa, submucosa, circular muscle, and longitudinal muscle layers. Measurements were conducted blindly by the same person using NIS elements software. For each sample, the layer thickness was measured at twelve different locations and averaged (three measurements per photo). When possible, measurements were only taken from images where the entire intestinal wall could be observed. Additionally, Masson’s trichrome staining was employed to detect fibrosis in the gastric and intestinal tissues.

### 2.5. Statistical Analysis

The GraphPad Prism^©^ 8.1.2 software was used for statistical analysis of the data. The unpaired Student’s *t*-test was used for comparison between the two groups (diabetic and control) since the variables had a Gaussian distribution. The Shapiro–Wilk test was applied to assess the normality of data distribution. Data were expressed as mean ± SEM, percentage (%), or median, as appropriate, while “*n*” refers to the number of cats in each group. The two-way ANOVA followed by an unpaired *t*-test with Welch’s correction was used in the data from the histopathological evaluation. For ultrasound measurements, comparisons were made between the diabetic group and published reference values using unpaired Welch’s *t*-tests based on available literature-derived summary statistics (mean, standard deviation, and sample size). In all cases, a *p* value of less than 0.05 was considered to denote a statistically significant difference.

## 3. Results

### 3.1. Study Population

Of the 13 cats that participated in the ultrasound study, four were females and nine were males; all were sterilized. In terms of breed, all but two were European Shorthair, with the exceptions being a Siamese cat and a Norwegian Forest cat. The average age was 12.5 ± 1.17 years (range: 7–19 years), and the average weight was 5.61 ± 0.65 kg (range: 2.75–9 kg). As expected, among the 13 cats, only 1 was underweight (score 3), and 4 had a normal body condition (score 5), while 3 were overweight (scores 6 and 7), and the remaining 5 were obese (scores 8 and 9), according to the WSAVA body condition scoring system for cats [35]. The duration since diabetes diagnosis ranged from 7 days to 5 years, with a median of 2 months. The average blood glucose level measured before the ultrasound was 371.56 ± 45.99 mg/dL, with a range of 170 to 600 mg/dL. The upper limit of 600 mg/dL corresponds to the maximum reading capability of the glucometer used. This cat with this value had uncontrolled diabetes and was euthanized a few days after the ultrasound.

As expected, all cats were receiving treatment to control diabetes. Caninsulin^®^ and Lantus^®^ were the most used insulins (5 cats each), followed by Prozinc^®^ (1 cat) and Degludec^®^ (1 cat). Interestingly, only one cat was receiving a non-insulin treatment, which involved the administration of metformin. Only one cat was also receiving treatment not directed at diabetes, which was Impromune^®^, since this cat was positive to Feline Immunodeficiency Virus (FIV). Additionally, two other cats had health issues besides diabetes. One cat had chronic pancreatitis and was beginning to show signs of heart disease. The other cat was experiencing blindness and had degenerative lesions in the kidneys and liver. In both cases, no relationship was established between these other health issues and DM.

Regarding typical signs of diabetes, all but one owner reported the expected polydipsia and polyuria. However, only eight owners recognized polyphagia, while weight loss was observed in ten cats.

All demographic information related to the thirteen cats enrolled in this study and their typical diabetes signs are summarized in Table 1.

### 3.2. Owners’ Perception of Digestive Changes

Out of the thirteen owners that completed the anamnesis, eleven reported at least one digestive change in their diabetic animals, representing a prevalence of digestive alterations of 84.62%. Some cats exhibited either gastric or intestinal changes, but the majority experienced both.

Regarding the upper GI tract, six owners indicated that their cats went from not vomiting to consistently vomiting either around the time of diabetes diagnosis or afterward, with a related frequency of at least two to three times per week. Two of these owners noted that vomiting typically occurred within 30 min after a meal, and the cats maintained their appetite post-vomiting.

In terms of defecation habits, four owners reported an increase in defecation frequency, while seven reported an increase in stool volume. Only one owner reported constipation, with decreased defecation frequency. Diarrhea was described in seven animals, and tenesmus was noted in three. Six of the thirteen cats exhibited behavioral changes regarding defecation, starting to defecate outside the litter box, often on the floor. Among these six cats, three presented with diarrhea. One also exhibited an altered appetite, beginning to reject the usual solid foods, and another one started vocalizing during defecation. One owner specifically described that their cat nearly stopped using the litter box entirely for defecation.

Concerning fecal appearance, in addition to increased volume, owners reported various changes such as stronger odor (one cat); watery feces and yellowish color when defecating outside the litter box (one cat); darker color (one cat); and larger, thicker stools (one cat). In the open-ended section of the survey, one owner mentioned that their cat initially experienced constipation during the early months of diabetes, which subsequently evolved into diarrhea.

The main results related to owners’ perception of digestive changes are summarized in Figure 1.

### 3.3. Ultrasound Evaluation of the GI Tract

On average, compared to the maximum reference values (RVs) documented in the literature, cats exhibited increased thickness of the gastric rugal fold (5.58 ± 0.4 mm vs. RV: 4.22 ± 0.31 mm [34]) and inter-rugal (2.82 ± 0.08 mm vs. RV: 2.03 ± 0.41 mm [34]) (Figure 2) (*p* < 0.05). This increase in the thickness of the GI wall was also observed in the duodenum (3.19 ± 0.06 vs. RF: 2.20 ± 0.17 mm [32]) and jejunum (3.12 ± 0.12 vs. RF: 2.22 ± 0.18 mm [32]) (Figure 2) (*p* < 0.05 for both). On the other hand, the ileum and colon walls displayed normal thickness in diabetic cats (ileum: 3.21 ± 0.16 mm vs. RF: 3.00 ± 0.28 mm [32]; colon: 1.88 ± 0.15 vs. RF: 1.67 ± 0.20 mm [34]) (*p* > 0.05 for both).

Regarding the stomach, ultrasound measurements of the gastric wall at the level of the rugal fold ranged from 3.99 mm to 7.97 mm. Only one cat had average values within the normal reference range, while all the others presented values above the reference range. Three diabetic cats showed an average of the three measures above 6 mm (representative image in Figure 3A), which is typically considered pathological [32,33]. The inter-rugal thickness varied from 2.48 mm to 3.32 mm, and all the diabetic cats had average values from the three measurements above the reference range (Figure 3B).

In the duodenum (Figure 3C), values ranged from 2.98 to 3.55 mm, with all animals presenting values above the reference range. The same was true for the jejunum (Figure 3D), where values ranged from 2.63 to 3.66 mm.

The ileum was not easily visualized if filled with gas, meaning that it was not evaluated in all the diabetic cats. The veterinary radiologist was able to confidently measure the ileum in eight cats, with values ranging from 2.82 to 4.05 mm (Figure 3E). Three cats presented values above the reference value.

Although the average colon measurements did not differ from the normal reference values, the majority of the cats had values above 2 mm (Figure 3F). The measurements for the colon ranged from 1.24 to 2.62 mm.

In a normal GI ultrasound, five echogenic layers are identified: the innermost hyperechoic layer corresponds to the surface of the mucosa; the inner hypoechoic layer represents the mucosa; the middle hyperechoic layer is the submucosa; the outer hypoechoic layer is the muscularis propria; and the outermost hyperechoic layer is the subserosa/serosa [36]. Although there was sometimes an increase in overall wall thickness, normal GI mural stratification was preserved in all ultrasound examinations, allowing for clear identification of the previously described layers.

### 3.4. Necropsy and Histopathological Evaluation

Necropsies were performed on six diabetic cats and nine controls. Of the diabetic animals, only one was previously observed and submitted to an ultrasound examination by our research team. The remaining five were donated by other veterinary clinics. All the cats were European Shorthair, ranged from 10 to 14 years, and included four males and two females. Four of these animals were euthanized due to diabetic ketoacidosis, chronic kidney disease, and pulmonary failure, and two died spontaneously. None of these animals had a history of GI disease. Of the nine cats used as controls, six were euthanized due to various conditions: FIV (*n* = 1/9), pulmonary metastasis from mammary gland tumors (*n* = 1/9), high rise syndrome (*n* = 1/9), and renal failure due to chronic kidney disease (*n* = 3/9). The remaining three cats died spontaneously, without a determined cause of death. During necropsy, all control animals presented with an intact and healthy GI tract.

During necropsy, we found fecalomas in the colon and rectum of one diabetic cat and megaesophagus in another. Upon opening the intestinal segments, all the diabetic cats exhibited tense, turgid, firm, and thickened intestinal walls that remained curling upon intestine opening, instead of falling flaccid as expected (Figure 4A green arrow). Additionally, several areas of hyperemia diffusely distributed throughout various segments of the intestinal tract were observed in the mucosa, characterized by reddish patches with ill-defined boundaries (Figure 4A purple arrows), suggestive of inflammation.

Microscopic evaluation revealed that the GI wall was thickened in the stomach (3016.97 ± 486.20 µm vs. 2198.38 ± 75.58 µm, *p* = 0.0335), duodenum (2108.74 ± 175.27 µm vs. 1593.73 ± 68.28 µm, *p* = 0.0279), and jejunum (1781.49 ± 81.08 µm vs. 1239.89 ± 64.60 µm, *p* = 0.0007) of diabetic cats compared to controls (Figure 5A). This difference was not observed in the ileum (2409.85 ± 141.72 µm vs. 2111.52 ± 93.69 µm, *p* = 0.1561) or colon (1479.79 ± 163.23 µm vs. 1390.39 ± 111.40 µm, *p* = 0.6641) of diabetic cats compared to controls (Figure 5A).

Notably, the muscular layers were consistently increased across all studied sections in the diabetic cats compared to controls (gastric wall—longitudinal muscle: 273.18 ± 34.02 µm vs. 158.67 ± 11.34 µm, circular muscle: 1083.77 ± 237.35 µm vs. 483.25 ± 58.72 µm; duodenum–longitudinal muscle: 186.36 ± 17.99 µm vs. 126.54 ± 9.96 µm, circular muscle: 750.00 ± 90.48 µm vs. 315.77 ± 33.84 µm; jejunum–longitudinal muscle: 184.98 ± 16.41 µm vs. 92.93 ± 8.47 µm, circular muscle: 546.34 ± 52.49 µm vs. 250.11 ± 12.61 µm; ileum—longitudinal muscle: 305.00 ± 26.69 µm vs. 186.42 ± 19.39 µm, circular muscle: 800.46 ± 29.76 µm vs. 492.29 ± 28.74 µm; colon—longitudinal muscle: 323.53 ± 45.41 µm vs. 198.79 ± 23.84 µm, circular muscle: 237.38 ± 58.85 µm vs. 170.98 ± 20.68 µm, respectively, *p* < 0.05 for all). However, the mucosal layer showed a significant increase only in the jejunum of the diabetic cats (873.38 ± 25.96 µm) compared to control cats (737.23 ± 39.98 µm) (*p* = 0.0178).

Representative microscopic photographs of all the intestinal segments of control and diabetic cats stained with hematoxylin and eosin are shown in Figure 6.

In addition to the quantitative analyses, a qualitative assessment was performed by an experienced pathologist. Diabetic cats exhibited gut-associated lymphoid tissue (GALT) hyperplasia (Figure 6) and inflammatory infiltrate (Figure 7) throughout all sections of the GI tract, which were absent in control animals. The inflammatory infiltrates were predominantly lymphoplasmacytic, with occasional eosinophils (Figure 7B). These infiltrates were more pronounced in the small intestine but were also observed in the stomach and colon. They were primarily located within the mucosa, although a smaller number of inflammatory cells extended into the submucosa. In the small intestine, the infiltrate was so exuberant that it disrupted the normal mucosal architecture, leading to villous enlargement and increased spacing between crypts. Additionally, a marked accumulation of inflammatory cells was noted within intestinal vessels, consistent with leukocytosis (Figure 7D). The histological findings are characteristic of gastroenteritis.

Furthermore, Masson’s trichrome staining of the diabetic GI tract revealed abnormal collagen deposits across all intestinal segments studied, with a particularly pronounced accumulation in the muscular layers of diabetic cats. This staining technique differentiates collagen from other tissue components by coloring it blue, while muscle fibers appear red [37]. The intense blue patches observed within the muscular layers of diabetic cats indicate collagen deposition, suggesting fibrosis and structural remodeling within the intestinal wall. Representative images are shown in Figure 8.

## 4. Discussion

This study marks the first exploration into the GI health of diabetic cats, unveiling intriguing parallels with human diabetes. The findings suggest that diabetic cats may experience similar symptoms to those observed in humans, with 83% of cat owners reporting noticeable digestive changes. Ultrasound evaluations revealed significant thickening of the GI wall, while histopathological analysis uncovered widespread fibrosis and inflammatory infiltrates throughout the GI tract.

All demographic data of the cats enrolled in this study align with expectations. Most cases of spontaneous diabetes occur in middle-aged to older cats (10–14 years) [6], and the average age of the cats that underwent ultrasound falls within this range, as well as all the necropsied diabetic cats. The fact that most of the animals were neutered obese males is also consistent with existing literature [7]. Male cats are significantly more predisposed to diabetes compared to females due to gender differences in weight gain and insulin sensitivity [38]. Additionally, male cats are more prone to weight gain, are more negatively affected by it, and have higher basal insulin levels with lower insulin sensitivity [39]. Similar to findings in human medicine, obesity, together with physical inactivity, are believed to be the main contributors to the insulin resistance associated with diabetes in cats [38,40]. Therefore, it is understandable that more than 60% of the cats in this study were overweight or obese. Additionally, contrary to what is described in dogs [41], neutering is a risk factor because the cats become more prone to becoming overweight, as gonadectomy reduces energy requirements and increases voluntary food consumption [40].

Diabetic chronic hyperglycemia leads to elevated glucose levels in the glomerular filtrate, and the presence of unabsorbed glucose acts as an osmotic solute in urine, causing osmotic diuresis, polyuria, and thirst, resulting in increased water intake [42,43]. It is therefore unsurprising that all owners but one reported observing polydipsia and polyuria in their pets. In individuals with diabetes, despite high blood glucose levels, there is a lack of glucose uptake by the cells, leading to reduced body mass and weight loss [43], and polyphagia emerges as a compensatory response [44]. This explains simplistically why most cats also present with polyphagia and weight loss.

The glycemia values of cats included in this study indicate poor glycemic control. Most of the cases involved animals admitted to UPVet due to high glycemic episodes. Diabetic companion animals’ owners usually report difficulties in managing and administering treatment to their diabetic animals, which impact their daily routines and quality of life, representing not only a temporal but also a financial burden [45]. This challenge in maintaining proper treatment likely explains why only one cat in the study had been diagnosed with diabetes for five years and another for two years, while the remaining cats had been diagnosed with diabetes for only a few months or days. This aligns with the literature, which states that 1 in 10 cats is euthanized at the owner’s request at the time of diabetes diagnosis [46], and the mortality rate in diabetic cats within the first 3–4 weeks is 11–17% [47]. A more recent study found that the median survival time for diabetic cats was 516 days, with a range of 1 to 3468 days [48]. When considering euthanasia, owners reported that concurrent disease, costs, and age were the most important factors [45]. Hence, it makes sense to look at the possible impact of diabetes in the GI tract of diabetic cats. A study investigating the impact of GI complications in diabetic human patients found that these significantly decrease health-related quality of life, affecting not only physical functioning and general health perceptions but also vitality, social functioning, and emotional and mental health [18]. Interestingly, GI complications of diabetes appear to affect up to 75% of diabetic human patients [16,17], and this study suggests they may affect more than 80% of diabetic cats. Some of the most common GI symptoms of diabetes in human population are vomiting (mostly due to gastroparesis), constipation, diarrhea, and fecal incontinence [20,49]. Thus, it is not surprising that the most common digestive changes described by the owners of diabetic cats are vomiting and diarrhea.

Gastroparesis in diabetic human patients is extensively studied but remains poorly understood [20]. Poor glycemic control seems to be enough to cause disrupts in gastric coordination and emptying [20], and the presence of neuronal damage [50] and remodeling of the gastric wall [51] are also identified risk factors. Indeed, a decreased number and phenotypic changes of myenteric neurons [50], a decreased expression of nitrergic neurons, and reduced number of Interstitial Cells of Cajal [52] have been linked to gastroparesis and vomiting [53] in laboratory animals and humans. In companion animals, a single study evidenced a notable reduction in the density of nitrergic neurons in both the antrum and ileum of diabetic dogs compared to the control group [54].

One common observation in diabetic cats was also an increase in fecal excretion. This aligns with our own research, which also noted increased fecal excretion in STZ-induced diabetic animals. These findings may be attributed to polyphagia and intestinal distension [23]. Furthermore, diarrhea in diabetic patients is multifactorial and may involve the accumulation of advanced glycation end-products, neuronal damage, and remodeling of the intestinal wall, especially fibrosis of the muscular layers [24]. It is typically intermittent, watery, painless, nocturnal, and may be associated with fecal incontinence in at least a third of the patients [55,56]. The fact that almost half of the owners (6/13) reported that the cats started to defecate outside the litter box may indicate that these animals also suffer from fecal incontinence. In human patients, episodes of incontinence are considered a troublesome symptom and may be attributed to anal sphincter dysfunction and neuronal damage, potentially exacerbated by acute hyperglycemic episodes [57] that inhibit the sphincter and reduce rectal compliance [20,58]. Feline fecal incontinence usually suggests neurologic-related anal sphincter incontinence [59]. In diabetic patients, this symptom indicates poor glycemic control [57], raising the question of whether the suboptimal glycemic control observed in the cats in this study may also contribute to the alteration in defecation habits. Considering the burden of caring for diabetic cats on their owners, having the cats defecate outside the litter box can represent a significant additional challenge [60]. Discovering feces at home can be a significant source of frustration for cat owners as it is considered unpleasant and unhygienic to live with a pet with this condition [59,60]. This problem demands both time and financial investment in cleaning and possibly repairing surfaces, and neglecting to address this issue can strain the bond between human and animal [60]. In fact, house soiling is a major cause for cats being abandoned or euthanized [60,61]. Given that we observed this behavior in almost half of the cats that underwent ultrasound, the authors of this study believe it would be useful to distribute a general questionnaire to the owners of diabetic cats. This approach is essential to determine if this is, indeed, a common issue among diabetic cats.

While owners report various digestive changes in diabetic cats, it prompts the question: are there corresponding morphological changes in the GI tract of these animals? Ultrasonography has emerged as a cornerstone in diagnosing intestinal changes in cats [62], since most GI pathologies can alter the thickness and/or integrity of the intestinal wall layers [62,63,64]. In this study, we found that while the integrity of the intestinal layers is maintained, there is a thickening in the jejunum, duodenum, and stomach, with some animals exhibiting a gastric wall thicker than 6 mm. The histopathological results supported the ultrasound findings in diabetic cats, as increased thickness of the GI wall was consistently observed in the stomach, duodenum, and jejunum. However, morphometric analyses additionally revealed that the muscle layers in all studied sections were increased in diabetic cats compared to controls.

Previous studies have established that increased thickness of gastric muscle layers, due to collagen deposition, is common in both diabetic patients and experimental diabetic models (mostly rats) [65,66]. This increased thickness is responsible for greater stiffness, absorption disturbances, and abnormal motility of the gastric wall, potentially resulting in either faster or delayed gastric emptying, contributing to the gastric symptoms in diabetic patients [67] and possibly explaining the increased frequency of vomit in diabetic cats referred by almost half of diabetic cat owners. Additionally, food retention in the stomach combined with posterior accelerated gastric emptying contributes to poor post-prandial glycemic control, leading to irregular hyper and hypoglycemic episodes [68], which can also be related to the poor glycemic control observed in the diabetic cats in this study.

The intestinal thickening found in diabetic cats aligns with what has been extensively described by other researchers, including our own research group [23,24,69,70,71]. The increase in the thickness of mucosa seems to be related to increased food intake [50], increased expression in diabetic animals of glucagon-like peptide-2 that has a trophic action on the intestinal epithelium [72], and suppression of apoptosis [73]. The increased thickness in the muscle layers appears to be directly related to the accumulation of Advanced Glycation End Products [70] and collagen type I [74]. The collagen fibers accumulate mostly around and between smooth muscle cells, causing stiffening of the diabetic gut and decreased resting compliance. In addition to extracellular matrix remodeling, authors also found smooth muscle cell hypertrophy [74]. This remodeling is significant, as it can influence absorption and cause small intestinal bacterial overgrowth and motility disorders, contributing to symptoms such as constipation, diarrhea, and fecal incontinence [25,75], symptoms similar to those observed in the diabetic cats included in this study. The fact that GI wall thickening in diabetic cats is gradually less preeminent in the distal direction is also compatible with what was described by our own group [23] and Fregonesi et al. [50]. They showed that there is a differential effect of diabetes in the GI tract, with the distal segments being affected last [50].

In addition to intestinal remodeling and fibrosis, the histopathological results also revealed the presence of diffuse inflammatory infiltrate in all segments of the GI tract, mainly involving the mucosa and submucosa. These findings were not surprising, as inflammatory infiltrates have previously been observed in the stomach [76], intestine [77], and colon [78] of both diabetic patients and laboratory animals. Diabetes, particularly T2D, is often associated with chronic low-grade inflammation with an increase in circulating inflammatory cytokines. This systemic inflammation is linked to insulin resistance and can affect various organs, including the GI tract, by disrupting normal cellular functions and promoting inflammatory responses [79]. Additionally, diabetes can cause changes in the gut microbiota, leading to dysbiosis and the promotion of inflammation in the gut [78]. It is also associated with increased intestinal permeability (“leaky gut”), which allows endotoxins and inflammatory mediators to enter the bloodstream [80]. These inflammatory infiltrates are important as they can be associated with other GI alterations such as fibrosis and can impact gut function and further contribute to GI symptoms of diabetes [81].

This study presents some limitations that should be considered when interpreting the findings. One of the main constraints relates to the use of reference values from the literature for the interpretation of ultrasound data in the absence of an internal control group. While these values provide a useful benchmark, they cannot be directly correlated with the histopathological findings obtained. Additionally, the relatively small sample size, though acceptable for an exploratory study, may limit the statistical power and generalizability of the results. A larger sample size would be beneficial in future studies to enhance the robustness of the findings. Owner-reported clinical signs, gathered through semi-structured questionnaires, may also be subject to variability and interpretation bias. All diabetic cats in this study were reportedly fed commercial dry diabetic food, which likely helped reduce dietary variability. While these diets share similar nutritional goals, minor differences between brands and owner-reported data may still introduce some variability. The specific impact of such diets on gastrointestinal morphology is not well established and should be further explored in future studies with controlled feeding protocols. Lastly, the study focused primarily on structural and histological assessments, without integrating functional analyses such as gastrointestinal transit time or motility tests, blood biochemical markers, endotoxin levels, and alterations in gut microbiota, which could provide a more comprehensive view of the underlying pathophysiology.

## 5. Conclusions

This pioneering study is the first to investigate the GI health of diabetic cats, revealing significant findings that align with patterns seen in human diabetic patients. Remarkably, more than 80% of the diabetic cats in our study displayed at least one GI issue, with increased vomiting frequency, diarrhea, and defecation outside the litter box being common problems. Both ultrasound and histopathological evaluations uncovered notable thickening of the GI wall in the stomach, duodenum, and jejunum. Additionally, we observed increased thickness of the muscular layers throughout the entire length of the GI tract, accompanied by inflammatory infiltrate and fibrosis. These findings suggest that diabetic cats experience GI symptoms and intestinal remodeling like those observed in human patients and experimental models of diabetes.

This research underscores the significant impact of diabetes on feline digestive health, opening new avenues for understanding and treating this condition in pets. However, further research is essential to fully grasp how these GI changes affect the quality of life for both diabetic cats and their owners. It also highlights the importance for veterinarians to consider these potential alterations when developing treatment plans for diabetic cats.

## Figures and Tables

**Figure 1 vetsci-12-00529-f001:**
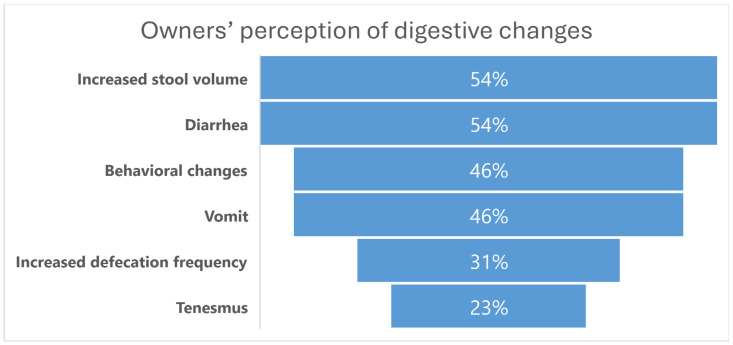
The percentage (%) of digestive alterations reported by owners of diabetic cats (*n* = 13). Behavioral changes primarily involved defecation outside the litter box.

**Figure 2 vetsci-12-00529-f002:**
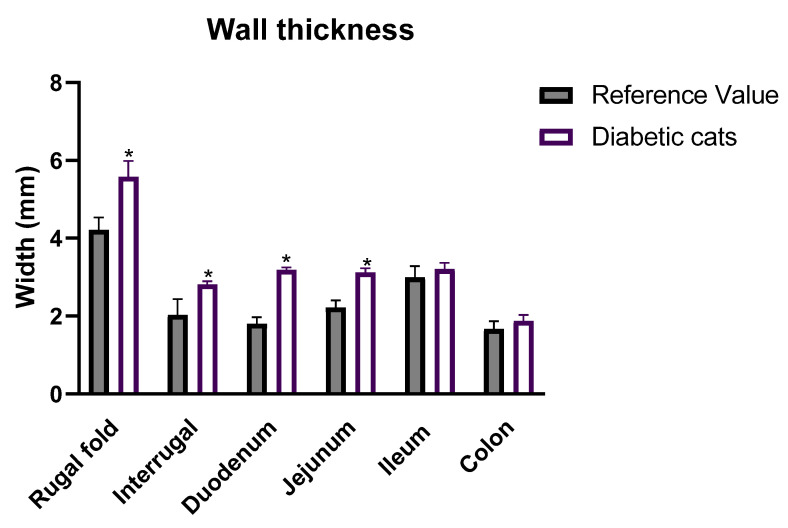
Total wall thickness (mm) of the gastric wall (rugal fold and inter-rugal), duodenum, jejunum, ileum, and colon measured using ultrasound in diabetic cats (*n =* 13) compared to reference values previously documented in the literature. Values are mean ± SEM, and unpaired Welch’s *t*-tests were used to compare the two experimental groups (control and diabetic cats). * Statistical difference *p* < 0.05 vs. correspondent control.

**Figure 3 vetsci-12-00529-f003:**
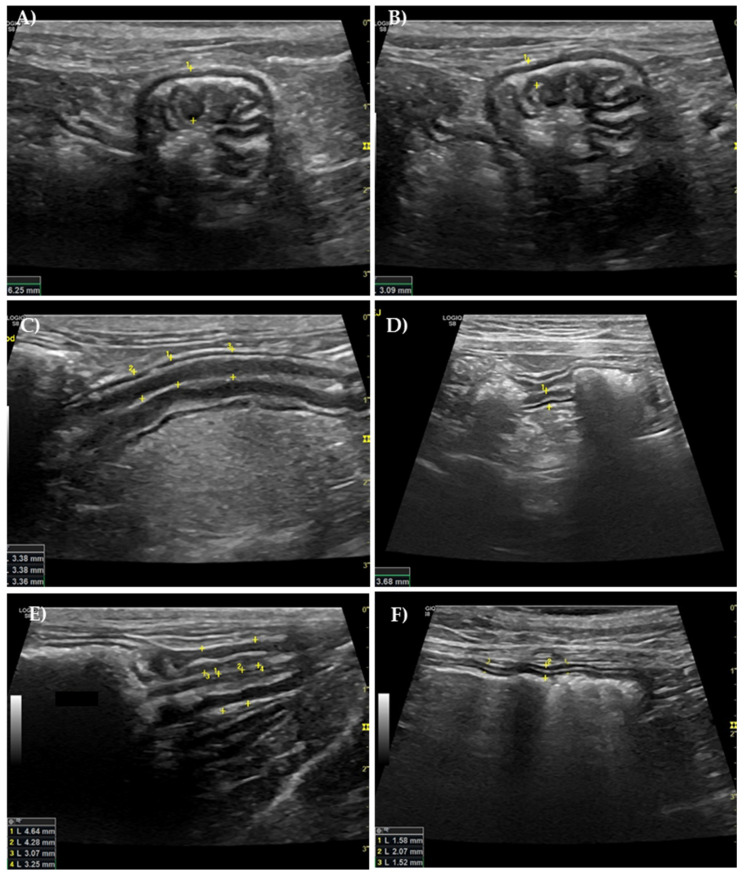
Representative ultrasound images of the gastric rugal fold (**A**) and inter-rugal (**B**), duodenum (**C**), jejunum (**D**), ileum (ileo-colic transition) (**E**), and colon (**F**) of diabetic cats, longitudinal plans, using a 9–11 MHz probe. In these images, all portions except the colon presented with wall thickening compared to the reference value (RV) (RV: gastric rugal fold = 4.22 mm; gastric inter-rugal = 2.03 mm; duodenum = 2.20 mm; jejunum = 2.22 mm; ileum = 3.20 mm; colon = 1.67 mm). The yellow numbers represent the number of measurements taken on each image, which are delimited by the "*+*" symbols.

**Figure 4 vetsci-12-00529-f004:**
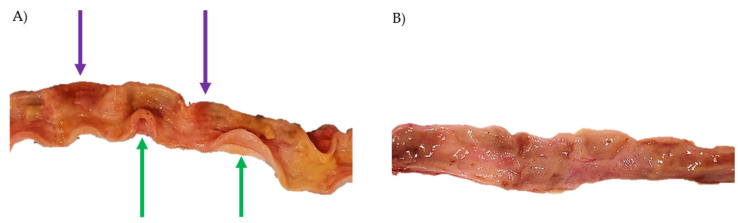
Photos taken during intestinal necropsy of (**A**) a diabetic cat—jejunal wall exhibited significant thickening, curling upon opening of the intestine (green arrow) rather than falling as expected, accompanied by areas of hyperemia (purple arrows); (**B**) control cat—normal jejunal wall.

**Figure 5 vetsci-12-00529-f005:**
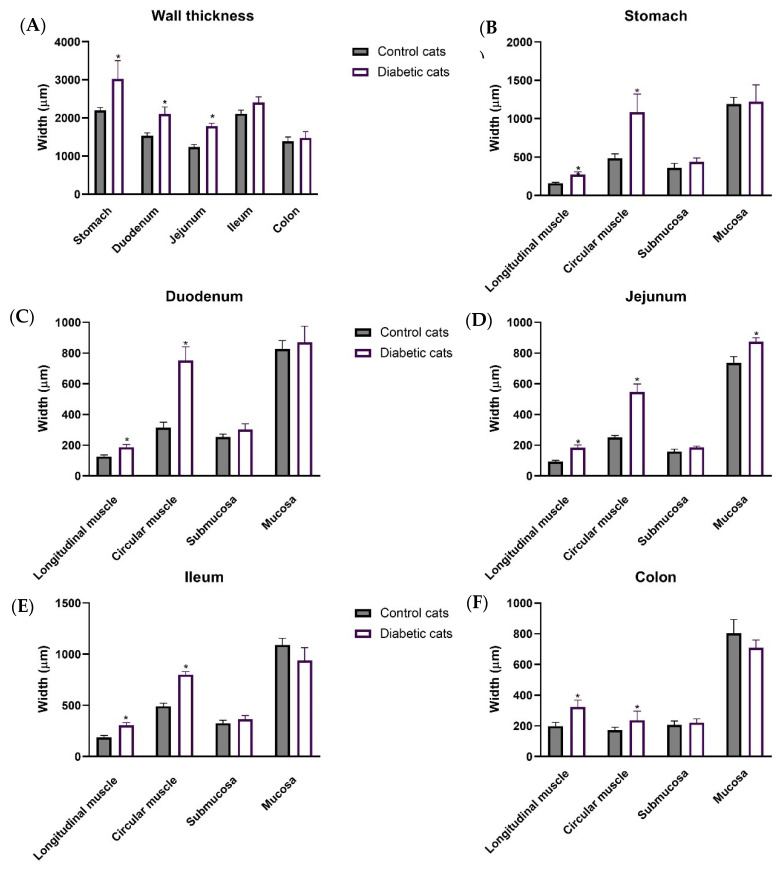
Morphometric evaluation of intestinal segments (stomach, duodenum, jejunum, ileum, and colon) of control cats (*n* = 9) and diabetic cats (*n* = 6). Total wall thickness (μm) of each intestinal segment (**A**); thickness (μm) of the intestinal layers (longitudinal muscle, circular muscle, submucosa, and mucosa) of stomach (**B**), duodenum (**C**), jejunum (**D**), ileum (**E**), and colon (**F**). Values are mean ± SEM and a 2-way ANOVA followed by an unpaired *t*-test with Welch’s correction was used to compare the two experimental groups (control and diabetic cats). * Statistical difference *p* < 0.05 vs. correspondent control.

**Figure 6 vetsci-12-00529-f006:**
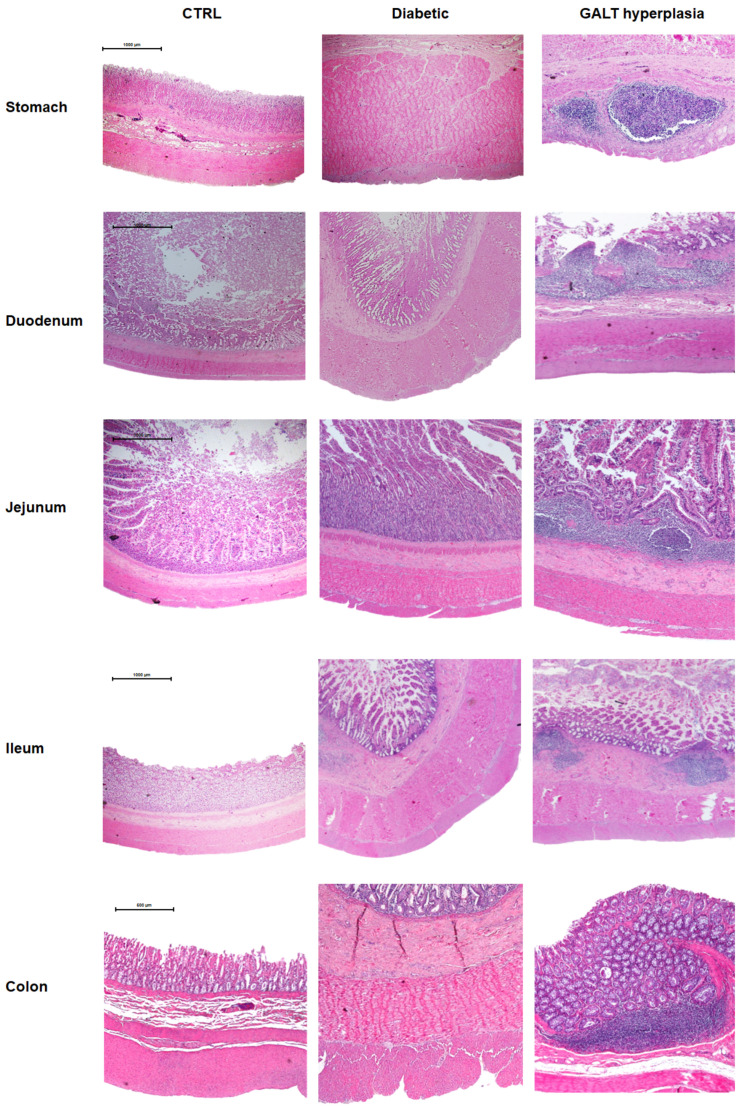
Representative microscopic images of all intestinal segments from control and diabetic cats, stained with hematoxylin and eosin, as well as gut-associated lymphoid tissue (GALT) hyperplasia observed throughout the gastrointestinal tract of diabetic cats. All images were captured at 20× magnification, except for the colon, which was captured at 40×. The scale bar in each image is valid to the entire line.

**Figure 7 vetsci-12-00529-f007:**
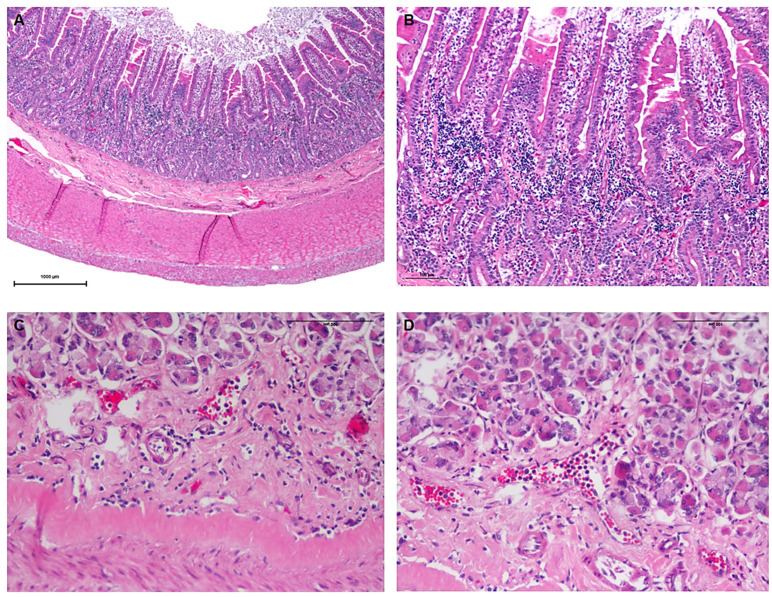
Representative microscopic images of inflammatory infiltrates in the gastrointestinal tract of diabetic animals. (**A**) Jejunum (20× magnification) showing prominent inflammatory infiltrates, particularly within the mucosa, with disruption of the normal architecture. (**B**) Higher magnification (100×) of the jejunum highlighting the abundance and cellular composition of the infiltrate, predominantly lymphoplasmacytic. (**C**) Stomach (200× magnification) displaying inflammatory cells in close proximity to the muscularis mucosae. (**D**) Stomach (200× magnification) showing marked leukocytosis with numerous inflammatory cells within blood vessels.

**Figure 8 vetsci-12-00529-f008:**
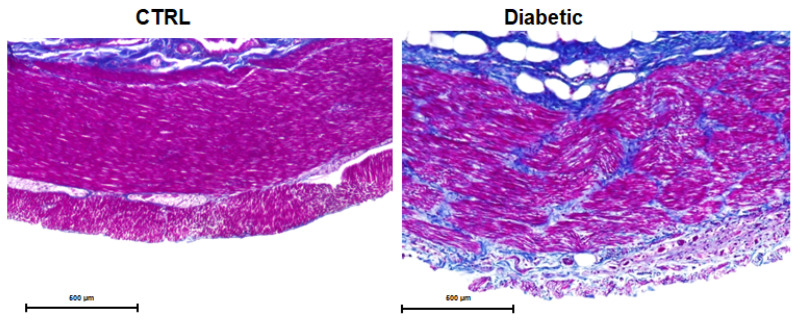
Representative microscopic images of jejunum of controls (CTRL) and diabetic cats, stained with Masson’s trichrome. The blue patches within the muscular layers of diabetic gut indicate collagen deposition. Images captured with a 40× magnification.

**Table 1 vetsci-12-00529-t001:** Basic animal identification data and typical diabetes signs are presented as mean ± SD, median, or percentage (%), as appropriate (*n* = 13 cats). FIV—Feline Immunodeficiency Virus.

	**Animal Data**
Gender	9 males (69.23%) 4 females (30.77%)
Age	12.5 ± 1.17 [7–19 years]
Weight	5.61 ± 0.65 [2.75–9 kg]
Body condition	Underweight (7.69%, *n* = 1) Normal (30.77%, *n* = 4) Overweight (23.07%, *n* = 3) Obese (38.46%, *n* = 5)
Time since diabetes diagnosis	2 months [7 days to 60 months]
Glycemia	371.56 ± 45.99 mg/dL [170–600 mg/dL]
Diabetes treatment	Caninsulin^®^ (38.46%, *n* = 5) Lantus^®^ (38.46%, *n* = 5) Prozinc^®^ (7.69%, *n* = 1) Degludec^®^ (7.69%, *n* = 1) Metformin^®^ (7.69%, *n* = 1)
Comorbidities	FIV (7.69%, *n* = 1) Chronic pancreatitis (7.69%, *n* = 1) Blindness (7.69%, *n* = 1) Kidney and liver degenerative disease (7.69%, *n* = 1)
Typical diabetes signs	Polydipsia (92.31%, *n* = 12) Polyuria (92.31%, *n* = 12) Polyphagia (61.54%, *n* = 8) Weight loss (76.92%, *n* = 10)

## Data Availability

Dataset available in this article.

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
