# Peer review of "Exploring Gastrointestinal Health in Diabetic Cats: Insights from Owner Surveys, Ultrasound, and Histopathological Analysis"

_vetsci, 2025, doi:10.3390/vetsci12060529_

Round 1
Reviewer 1 Report
Comments and Suggestions for Authors
In this manuscript, the authors studied the impact on stomach and bowel from thirteen type-2 diabetic cats, by using short surveys, imaging (ultrasound) and histopathology data. Thus, 83% of the owners reported some type of intestinal issue. The ultrasound imaging showed a increase of the thickness of stomach and intestinal wall, a fact that was corroborated with the histopathologic study (which includes necropsy data from 6 diabetic cats and 9 controls), which showed inflammatory infiltrate of mucosa/submucosa, an increase of thickness of inner muscular layer with fibrosis on this area. Based on these data, the authors conclude that type-2 diabetic cats experience similar GI alterations than those described for human and experimental models, facts that veterinary clinicians should be aware when developing therapeutic strategies for the treatment of this disease on cats.
The topic is interesting and has clinical relevance. Nevertheless, there are some minor issues which may be solved before considering this report to be published.
- Regarding the survey, was it previously validated? A validated questionnaire should increase the reliability of the data presented.
- Was performed any statistic test to compare ultrasound GI thickness between controls vs diabetic cats? As Fig. 2 and Fig. 5-A seem to have some correlation, similar results should be obtained.
- Regarding histopathology, although there is detailed quantitative data about the thickness of GI muscular layers, little is mentioned about the inflammatory infiltrate. On the basis on the representative images from stomach (which surprisingly seem to have only submucosal inflammation), duodenum, jejunum and ileon, it´s seem that diabetic cats should have extensive gastroenteritis (probably chronic), which may be one the cause of either GI disorders and the muscular hyperplasia and fibrosis observed. Thus, little information about this inflammatory infiltrate should be provided in the results section.
- Regarding the histopathology analysis, was detected any sign of diabetic intestinal neuropathy?
- Regarding the Masson´s trichrome analysis, is only mentioned that there is blue staining on all GI sections on diabetic cats. On the basis on fibrosis is an important histopathologic finding, more rigor is needed in the description of this finding. Therefore, a quantitative (or semi-quantitative) analysis should be formally presented.
- Figure 7. As it´s presented, the jejunum image seems to have a different magnification in comparison with the stomach (maybe 100X?). Additionally, similar representative images from control cats should be included.
- Lastly, it would be helpful to reflect the limitations of the study in order to provide to the readers some clues to further research topics.
Author Response
Dear Reviewer 1,
Thank you for the careful reading of our manuscript and for acknowledging the clinical relevance of our work. We appreciate the constructive comments and suggestions provided. Below, we address each point raised and detail the modifications made to improve the clarity and scientific rigor of the manuscript.
1. Regarding the survey, was it previously validated? A validated questionnaire should increase the reliability of the data presented.
Thank you for your comment, which is both appropriate and appreciated. The questions posed to the cat owners were adapted from semi-structured questionnaires commonly employed in veterinary clinical practice to aid in the investigation of gastrointestinal disturbances. These instruments are informed by well-documented clinical signs—such as vomiting frequency, stool consistency, appetite fluctuations, and weight changes—which are widely cited in the literature as reliable indicators of gastrointestinal dysfunction in feline patients. By focusing on these observable and owner-reported parameters, this approach provides a non-invasive yet clinically meaningful means of characterizing gastrointestinal alterations. This information was added to the manuscript for more clarity (blue). Nonetheless, your observation highlights an important consideration. As we plan to expand our investigation into gastrointestinal complications in cats—particularly by developing an online form for diabetic cat owners—we recognize the need to design a questionnaire that is both scientifically rigorous and user-friendly. Ensuring clarity and minimizing ambiguity in the phrasing of questions will be essential to obtaining reliable and reproducible data.
2. Was performed any statistic test to compare ultrasound GI thickness between controls vs diabetic cats? As Fig. 2 and Fig. 5-A seem to have some correlation, similar results should be obtained.
Thank you for your thoughtful observation. Yes, statistical comparisons of GI wall thickness obtained by ultrasonography were performed between diabetic and control cats using unpaired Welch’s t-tests and this information was not in the manuscript and was now added (blue) and the graph was updated. These tests were selected to account for comparisons with literature-derived reference values, as individual raw data for the control group were not available for ultrasound measurements.
Regarding your observation on Figure 2 (ultrasound) and Figure 5-A (histopathology), we agree that there appears to be a visual correlation in the pattern of GI thickening between modalities. However, while histopathological measurements (Figure 5-A) were directly compared between diabetic and control groups using 2-way ANOVA (based on our own data), the ultrasound measurements (Figure 2) were compared to published reference values. Therefore, although the trends are similar, the statistical framework differs due to the nature of the available data. This distinction will be clarified in the revised manuscript to avoid confusion. Thank you for highlighting the importance of consistency in interpretation across modalities.
3. Regarding histopathology, although there is detailed quantitative data about the thickness of GI muscular layers, little is mentioned about the inflammatory infiltrate. On the basis on the representative images from stomach (which surprisingly seem to have only submucosal inflammation), duodenum, jejunum and ileon, it´s seem that diabetic cats should have extensive gastroenteritis (probably chronic), which may be one the cause of either GI disorders and the muscular hyperplasia and fibrosis observed. Thus, little information about this inflammatory infiltrate should be provided in the results section.
Thank you for your thoughtful observation. You are absolutely right — although the primary focus of our study was on the morphometric evaluation of the gastrointestinal (GI) tract, the extent of the inflammatory infiltrates indeed warrants more detailed description. Taking your comment into consideration, we have now included a comprehensive histological description of the inflammatory infiltrates observed throughout the GI tract as marked in blue.
“In addition to the quantitative analyses, a qualitative assessment was performed by an experienced pathologist. Diabetic cats exhibited Gut-associated lymphoid tissue (GALT) hyperplasia (figure 6) and inflammatory infiltrate (figure 7) throughout all sections of the GI tract, which were absent in control animals. The inflammatory infiltrates were predominantly lymphoplasmacytic, with occasional eosinophils (figure 7 B). These infiltrates were more pronounced in the small intestine but were also observed in the stomach and colon. They were primarily located within the mucosa, although a smaller number of inflammatory cells extended into the submucosa. In the small intestine, the infiltrate was so exuberant that it disrupted the normal mucosal architecture, leading to villous enlargement and increased spacing between crypts. Additionally, a marked accumulation of inflammatory cells was noted within intestinal vessels, consistent with leukocytosis (figure 7 D). The histological findings are characteristic of gastroenteritis.”
Additionally, we have added higher-magnification images (figure 7) to better illustrate the cellular composition of the infiltrates. This added information helps to support the interpretation of the lesions as part of a chronic gastroenteritis, which, as you suggest, may contribute to the muscular alterations identified.
4. Regarding the histopathology analysis, was detected any sign of diabetic intestinal neuropathy?
Thank you for raising this important point. Although the detection of diabetic intestinal neuropathy would indeed be highly relevant, in our study this analysis was limited by technical constraints. Specifically, the animals were not always immediately refrigerated after death, and in some cases we observed varying degrees of autolysis, which compromised the quality of the tissue — particularly for evaluating fine neural structures. As a result, we were not able to reliably assess neuropathic changes using the available histological images alone. We agree that the investigation of diabetic enteric neuropathy is of great interest, and we are considering the use of specific neuronal and glial markers in future studies to address this question more directly and accurately.
5. Regarding the Masson´s trichrome analysis, is only mentioned that there is blue staining on all GI sections on diabetic cats. On the basis on fibrosis is an important histopathologic finding, more rigor is needed in the description of this finding. Therefore, a quantitative (or semi-quantitative) analysis should be formally presented.
Thank you for this valuable comment. In this study, the evaluation of fibrosis using Masson’s trichrome staining was qualitative, based on the presence or absence of blue staining. The difference was quite evident, as is now clearly illustrated with the addition of a representative control image. That said, we fully agree that a more rigorous, quantitative or semi-quantitative assessment would be especially valuable — particularly in the context of testing therapeutic interventions. Due to time constraints and the exploratory nature of this work, such an analysis was not included in the current version. However, we recognize its importance and are considering it for future studies where fibrosis will be assessed in a more systematic and quantitative manner.
6. Figure 7. As it´s presented, the jejunum image seems to have a different magnification in comparison with the stomach (maybe 100X?). Additionally, similar representative images from control cats should be included.
Thank you for pointing that out. We chose to include both control and diabetic cat images to allow for clearer comparison and improved clarity in the interpretation of the findings.
7. Lastly, it would be helpful to reflect the limitations of the study in order to provide to the readers some clues to further research topics.
Thank you for your observation. The limitations of the study were added:
“This study presents some limitations that should be considered when interpreting the findings. One of the main constraints relates to the use of reference values from the literature for the interpretation of ultrasound data, in the absence of an internal control group. While these values provide a useful benchmark, they may not be fully comparable to our study population and, importantly, cannot be directly correlated with the histopathological findings obtained. Additionally, the relatively small sample size, though acceptable for an exploratory study, may limit the statistical power and generalizability of the results. A larger sample size would be beneficial in future studies to enhance the robustness of the findings. Owner-reported clinical signs, gathered through semi-structured questionnaires, may also be subject to variability and interpretation bias. Lastly, the study focused primarily on structural and histological assessments, without integrating functional analyses such as gastrointestinal transit time or motility tests, blood biochemical markers and alterations in gut microbiota which could provide a more comprehensive view of the underlying pathophysiology.”
Reviewer 2 Report
Comments and Suggestions for Authors
I have several inquiries and suggestions for the author:
- Could you clarify the meaning of "Line 41 US evaluations"?
- In Lines 152-153, please specify the manufacturer and model of the General Electric Logiq S8 ultrasound machine used.
- For Table 1, I recommend revising the format to align with the standards of published articles in the journal.
- The text on the vertical axis of Figure 1 is not fully visible.
- In Figure 2, only the thickness of the intestinal wall is measured. Consider integrating this with Figure 5 for a more comprehensive analysis.
- There are additional metrics for assessing changes in gut health among obese individuals, such as blood biochemical markers, endotoxin levels, and alterations in gut microbiota. I suggest further exploration of these aspects to enhance the paper.
- The results depicted in Figure 6 appear to be captured at varying magnifications across groups. Please reorganize and present the results consistently, ensuring that the scale is clearly labeled.
- Why is there an absence of comparison between the control group and the normal group results, given that Figure 7 exclusively presents data for diabetic cats?
- In the results section, the explanation provided in the results section is excessively redundant and requires conciseness.
Author Response
Dear Reviewer 2,
We greatly appreciate your thoughtful and insightful comments, as well as the constructive suggestions you provided. Your feedback has played a significant role in enhancing both the clarity and overall quality of our manuscript. We have carefully considered each of your points, and in the following responses we address them individually, outlining the revisions made to improve the manuscript based on your input.
1. Could you clarify the meaning of "Line 41 US evaluations"?
Thank you for your comment. The phrase was reformulated in order to improve clarity (green).
2. In Lines 152-153, please specify the manufacturer and model of the General Electric Logiq S8 ultrasound machine used.
Thank you for your comment. The information was added as suggested (green).
3. For Table 1, I recommend revising the format to align with the standards of published articles in the journal.
Thank you for your comment. The table has now been updated accordingly.
4. The text on the vertical axis of Figure 1 is not fully visible.
Thank you for your comment. Indeed, some information was missing, but the image has now been corrected.
5. In Figure 2, only the thickness of the intestinal wall is measured. Consider integrating this with Figure 5 for a more comprehensive analysis.
Thank you for your observation. Regarding your observation on Figure 2 (ultrasound) and Figure 5 (histopathology), we agree that there appears to be a visual correlation in the pattern of GI thickening between modalities. However, while histopathological measurements (Figure 5) were directly compared between diabetic and control groups using 2-way ANOVA (based on our own data), the ultrasound measurements (Figure 2) were compared to published reference values. Therefore, although the trends are similar, the statistical framework differs due to the nature of the available data. This distinction will be clarified in the revised manuscript to avoid confusion. Thank you for highlighting the importance of consistency in interpretation across modalities.
6. There are additional metrics for assessing changes in gut health among obese individuals, such as blood biochemical markers, endotoxin levels, and alterations in gut microbiota. I suggest further exploration of these aspects to enhance the paper.
Thank you for your valuable comment. We appreciate the reviewer’s suggestion regarding the inclusion of additional metrics such as biochemical markers, endotoxin levels, and gut microbiota composition. We agree that these parameters can provide important insights into gastrointestinal health. However, the design and scope of the present study were specifically oriented toward the morphological characterization of gastrointestinal alterations and the assessment of clinical signs in diabetic cats. Moreover, due to the retrospective nature of part of the data and the fact that several animals had passed away by the time data consolidation was possible, it was not feasible to collect additional samples or access further biological material. We acknowledge the absence of functional and microbiological data as a limitation of the study, and this is explicitly stated in the revised manuscript. While these aspects were beyond the scope of our current objectives, we believe the combination of histopathological and imaging findings provides a valuable contribution to the understanding of gastrointestinal involvement in feline diabetes. Future prospective studies will certainly benefit from a more integrative approach, including biochemical and microbiota analyses.
7. The results depicted in Figure 6 appear to be captured at varying magnifications across groups. Please reorganize and present the results consistently, ensuring that the scale is clearly labeled.
Thank you for your comment. All images were acquired using 20× magnification, with the scale bar in each image corresponding to the entire line. The only exception is the colon images, which were captured at 40× magnification. This information has now been clarified in the revised figure legend.
8. Why is there an absence of comparison between the control group and the normal group results, given that Figure 7 exclusively presents data for diabetic cats?
Thank you for pointing that out. We chose to include both control and diabetic cat images to allow for clearer comparison and improved clarity in the interpretation of the findings.
9. In the results section, the explanation provided in the results section is excessively redundant and requires conciseness.
Thank you for your observation. With the alterations introduced in the Results section we hope that readers will no longer perceive the explanation as excessively repetitive and will find the section more concise and focused.
Reviewer 3 Report
Comments and Suggestions for Authors
The manuscript “Exploring Gastrointestinal Health in Diabetic Cats: Insights from Owner Surveys, Ultrasound, and Histopathological Analysis”, by Marisa Esteves-Monteiro, is a notable contribution to the knowledge of gastrointestinal pathophysiology in diabetic cats, possibly one of the least known aspects of the disease. The authors had access to analyze the digestive system of sterilized diabetic cats with diabetes but no previous digestive diseases. The authors observed signs of inflammation and fibrosis in these animals, suggesting non-clinical pathology in cats with diabetes.
Overall, the study is well conducted, offers new data hitherto unknown, and is of interest to the general reader, veterinarians, and cat owners. However, in the opinion of this reviewer, some modifications or clarifications must be made before the work can be accepted for publication. Especially the parallels between feline and human diabetes should be made with great caution.
Abstract
1.- The term stomach appears several times (in this section and in others of the manuscript) and it is more correct to use the term gastric.
2.- It would be better to avoid, in this section, comparisons with digestive pathology in diabetic humans, since the aim of the study is not to establish the cat as an animal model.
Introduction
1.- The work is important per se and should not make permanent references to diabetes in humans; among other things because they are global data and do not differentiate between type I and type II diabetes. All the data concerning humans in the Introduction should be in the Discussion. Consequently, the Introduction should be shortened and focus exclusively on the objective of the work.
2.- When you say, "spontaneous diabetes", do you mean the equivalent of type I or type II?
Material and Methods
1.- Well-structured and appropriate techniques. However, I consider it risk to compare data from animals that were diabetic with a very different time range (from 7 days to 5 years); It is likely that signs of inflammation predominate in the former and in the latter of hypertrophy of the muscle layers and fibrosis.
2.- The diabetic cats studied had been sterilized; Could sterilization be a co-cause of the observed effects? Could the authors provide some data on unsterilized diabetic cats? Was the cats' diet similar? This could be a key factor!
Results
1.- The clinical symptoms of the animals are very well described, although of course not all animals reproduced the same signs.
2.- Some reference should be made to the signs that predominate in relation to the time of evolution of the disease. Presumably, in cats with a recent diagnosis, signs of inflammation are likely to predominate, and in the latter, signs of hypertrophy of the muscle layers and fibrosis are predominant.
3.- The high quality of the ultrasound and structural studies should be highlighted.
I encourage the authors to make these slight modifications to this interesting manuscript.
Author Response
Dear Reviewer 3,
We would like to sincerely thank Reviewer 3 for the thorough and encouraging evaluation of our manuscript. We truly appreciate the recognition of the study's relevance and novelty, as well as the constructive comments provided. We have carefully considered all suggestions and concerns raised, particularly the important note regarding the parallels drawn between feline and human diabetes, which we have addressed with appropriate caution in the revised version. Below, we provide detailed responses to each of the reviewer’s points.
Abstract
1.- The term stomach appears several times (in this section and in others of the manuscript) and it is more correct to use the term gastric.
Thank you for your observation. The term “stomach” was intentionally retained in the Simple Summary to maintain accessibility for a broader audience. However, the term has been revised to “gastric” throughout the main text (highlighted in grey) to ensure scientific precision and consistency.
2.- It would be better to avoid, in this section, comparisons with digestive pathology in diabetic humans, since the aim of the study is not to establish the cat as an animal model.
Thank you for this pertinent observation. We agree that the aim of the study is not to establish the cat as an experimental model for human diabetes. However, due to the scarcity of data regarding gastrointestinal alterations in diabetic cats, references to human diabetic pathology were used to highlight the clinical relevance of such complications and justify the rationale for exploring this topic in feline medicine. Our intention is not to extrapolate findings from one species to another, but rather to underscore the potentially overlooked burden of gastrointestinal disease in diabetic cats, using existing human literature as contextual background. To address your concern, we have revised the abstract and main text (grey) to clarify this distinction and avoid any implication of translational modelling.
Introduction
1.- The work is important per se and should not make permanent references to diabetes in humans; among other things because they are global data and do not differentiate between type I and type II diabetes. All the data concerning humans in the Introduction should be in the Discussion. Consequently, the Introduction should be shortened and focus exclusively on the objective of the work.
Thank you for your thoughtful comment. We appreciate your recognition of the study’s intrinsic value. We agree that the aim of our research is not to model human diabetes, and that the focus should remain on feline patients. The references to human data were included as contextual background to underscore the clinical relevance of gastrointestinal (GI) complications in diabetic individuals and highlight the current knowledge gap in veterinary medicine. However, we understand your concern regarding the excessive emphasis on human diabetes in the Introduction. In response, we have revised and shortened the Introduction, keeping most of the human-related data and references in the Discussion section, where they are used to interpret our findings in light of what is known in other species.
2.- When you say, "spontaneous diabetes", do you mean the equivalent of type I or type II?
Thank you for this question. When referring to "spontaneous diabetes" in cats, our intention was to describe naturally occurring diabetes mellitus that arises without experimental induction and is not secondary to another disease directly affecting the pancreas (e.g., pancreatic neoplasia). In most cases, this form of diabetes resembles type 2 diabetes in humans, being characterized by insulin resistance and progressive β-cell dysfunction. However, feline diabetes is a heterogeneous condition, and a minority of cases may display features more consistent with type 1 diabetes. That said, we agree that the term "spontaneous" is not essential in this context and may lead to confusion. For clarity, we have removed it from the text in the revised manuscript.
Material and Methods
1.- Well-structured and appropriate techniques. However, I consider it risk to compare data from animals that were diabetic with a very different time range (from 7 days to 5 years); It is likely that signs of inflammation predominate in the former and in the latter of hypertrophy of the muscle layers and fibrosis.
Thank you for this important observation. We agree that the duration of diabetes can influence the nature and severity of gastrointestinal changes. In our sample, however, only one animal had been diagnosed with diabetes for five years; all the others had been diagnosed within a range of 7 days to 6 months. Notably, the cat with the longest duration did not present ultrasound and histological differences that were markedly distinct from those observed in animals with more recent diagnoses.
2.- The diabetic cats studied had been sterilized; Could sterilization be a co-cause of the observed effects? Could the authors provide some data on unsterilized diabetic cats? Was the cats' diet similar? This could be a key factor!
Thank you for raising this important point. All diabetic cats included in the study were sterilized, which reflects the typical profile of the diabetic feline population, as sterilized cats are more frequently diagnosed with diabetes. While sterilization is considered a risk factor—mainly through its association with weight gain and insulin resistance—it is unlikely to be a direct cause of the gastrointestinal alterations described in this study. Instead, it likely contributes indirectly by increasing susceptibility to diabetes and its metabolic consequences. Unfortunately, we did not have access to a any unsterilized diabetic cat to allow for meaningful comparisons.
Regarding diet, all animals were reportedly fed commercial dry diabetic food as recommended by their veterinarians. Although there may be slight variation between brands, the consistency in the type and formulation of the diets suggests that dietary differences were likely minimal. Nevertheless, as this information was owner-reported and not standardized across a controlled setting, we acknowledge this as a potential limitation, now addressed in the revised manuscript.
Results
1.- The clinical symptoms of the animals are very well described, although of course not all animals reproduced the same signs.
Thank you for your positive comment. Indeed, while the clinical symptoms were consistently assessed across animals, some degree of variability was expected, as not all individuals expressed the same signs.
2.- Some reference should be made to the signs that predominate in relation to the time of evolution of the disease. Presumably, in cats with a recent diagnosis, signs of inflammation are likely to predominate, and in the latter, signs of hypertrophy of the muscle layers and fibrosis are predominant.
Thank you for this insightful observation. While we acknowledge that the duration of diabetes may influence the nature of gastrointestinal alterations, our sample included mostly cats diagnosed within the previous 7 days to 6 months, with only one cat having a diagnosis of five years. Interestingly, this long-term diabetic cat did not show substantial differences compared to the more recently diagnosed animals. The only case in which a temporal progression of clinical signs was reported came from an owner who, in the open-ended section of the survey, described a transition from constipation in the initial months following diagnosis to diarrhea later on (lines 244-246).
3.- The high quality of the ultrasound and structural studies should be highlighted.
Thank you very much for your kind words. We are pleased that the quality of the ultrasound and structural analyses was appreciated.
Round 2
Reviewer 1 Report
Comments and Suggestions for Authors
All my queries have been considered and solved. Thank you for considering them. I have no more concerns.
Reviewer 2 Report
Comments and Suggestions for Authors
no comments
Reviewer 3 Report
Comments and Suggestions for Authors
The authors have satisfactorily answered the questions raised and therefore the manuscript can be accepted for publication